# Drug Shelf Life and Release Limits Estimation Based on Manufacturing Process Capability

**DOI:** 10.3390/pharmaceutics15041070

**Published:** 2023-03-26

**Authors:** Alexis Oliva, Matías Llabrés

**Affiliations:** 1Departamento de Ingeniería Química y Tecnología Farmacéutica, Facultad de Farmacia, Universidad de La Laguna, 38200 Tenerife, Spain; 2Catedrático Retirado de Farmacia y Tecnología Farmacéutica, Universidad de La Laguna, 38200 Tenerife, Spain; mllabres@ull.edu.es

**Keywords:** stability study, insulin, uncertainty, HPLC, release limits, specification limits, expiry time, process capability

## Abstract

Specification limits are the competence regulatory agencies, whereas the release limit is a manufacturer’s internal specification to be applied at the time of batch release to assure that quality attributes will remain within the specification limits until the expiry time. The aim of this work is to propose a method to set the shelf life from drug manufacture process capacity and degradation rate, using a modified version of the proposed method by Allen et al. (1991) Two different data sets were used to do this. The first data set corresponds to analytical method validation to measure the insulin concentration in order to estimate the specification limits, whereas the latter set gathered information on stability data of six batches of human insulin pharmaceutical preparation. In this context, the six batches were divided into two groups: Group 1 (batches 1, 2, and 4) was used to estimate shelf life; Group 2 (batches 3, 5, and 6) was used to test the estimated lower release limit (LRL). The ASTM E2709-12 approach was applied to verify that the future batches fulfill the release criterium. The procedure has been implemented in R-code.

## 1. Introduction

The planning and statistical data analysis of drug product stability is a controversial issue because the main goal of such studies is “to establish, based on testing a minimum of three batches of the drug substance or product, a retest period or shelf life and label storage instructions applicable to all future batches manufactured and packaged under similar circumstances” (ICH Q1E, Section 2.1) [1]. This procedure involves the estimation of two interrelated parameters, shelf life and release limits (RL) [2]. For a drug product containing a drug which degrades over time, ICH Q1E sets the shelf life of a drug product as the interception of the lower limit of the one-sided 95% confidence interval of the predicted drug concentration with the lower specification limit (LSL). This interception, which defines the shelf life (T), depends not only on the degradation rate but also on the drug concentration at time zero. Thus, lower and upper release intervals, [LRL, URL] located inside the lower and upper specification interval, [LSL, USL] must be set to provide assurance that the drug concentration will be within the specification limits until drug product’s expiring time. However, the condition for the release of a new batch, namely, that a sample taken from the new batch is inside the release interval, y ∈ [LRL, URL], y being an observation from a future batch, depends on the process capability to keep the manufacture under statistical quality control, in such a way that the drug content specification limits interval is not outside the LRL and URL.

It is clear that experimental designs and data statistical analysis commonly used in drug product stability trials are not suitable to obtain a release interval, and that these goals are unreachable outside the scope of quality by design (QbD) and pharmaceutical quality system paradigms [3]. Pharmaceutical QbD is a systematic approach to development that starts with predefined objectives and emphasizes the understanding and control of the product and the process on the basis of sound science and quality risk management [4]. One of the goals of pharmaceutical QbD is to increase process capability and reduce product variability, which often leads to product defects, rejects, and recalls. Achieving this goal requires a robust product and process design. Furthermore, a better understanding of the product and process may facilitate the identification and control of factors that affect drug quality. Finally, product and process capability is assessed and continuously improved after approval as part of product life cycle management. 

The main issues regarding experimental design and statistical models applied in establishing the expiry time have already been revisited [2,5]. In this respect, the authors have only one criticism, which is the excess of simulated data against real data; although simulation is a good tool to analyze and learn theoretical models, it is of limited use for model comparisons. 

Allen et al. [6] proposed the first method to set the RL, called the ADG approach, which seems to be the most prevalent one [7]. Since 1991, several methods have been proposed that can be classified as Frequentist [6,7,8,9] and Bayesian approaches [5,10,11,12,13]. The method described by Allen et al. (3) allows calculation of RL of potency of any dosage form and applies to any parameter for which the rate of change with time is predictably uniform and linear. Assurance is then provided at a 95% confidence level that any batch showing a mean release assay result within the calculated RL will remain within a registered limit at any subsequent time within its shelf life. However, their approach lacks rigorous statistical justification and does not incorporate all the information contained in intermediate samples. Wei [9] proposed three alternative methods for the determination of RL on the bases of potency assay results, known assay variability, and known batch-to-batch variability. The basic idea is to determine RL based on a strong correlation between the sample means of potency assay at time zero and those of at the end of shelf life, so that the chance of failing low release limit (LRL) will be controlled by selecting an appropriate RL. However, all three must know certain parameters to get started. Shao and Chow [10], Manola [11], and Hartvig [12] have described Bayesian approaches to setting release limits. These papers do not deal with the calculation of shelf life limits and assume that shelf life limits are already in place [9]. Yu et al. [13] proposed a Bayesian approach for determining the RL for critical quality attributes to keep the balance of both types of risks (i.e., the consumer’s risk of releasing a bad batch to the market and the manufacturer’s risk of rejecting a good batch) in the situation of multiple process steps and storage conditions. The proposed method can be used to control the overall risk of a multistep drug release process, as well as the risk of intermediate steps.

The aim of this work is to propose a method to set the release limits based on the manufacturing process capability and a modified interpretation of the method proposed by Allen et al. [6] The proposed procedure involves five steps: firstly, to verify that the manufacturing process is under statistical quality control and that the process capacity is high enough to leave room for drug degradation; secondly, the estimation of the lower and upper statistical control limits applying the general criteria μ±3σ; thirdly, estimation of stability data; fourthly, setting expiry time by a tradeoff of process capability and degradation rate using a modified version of the method already proposed by Allen et al. [6]; fifthly, to apply the ASTM E2709-12 method to verify that the future batches comply with the release criteria. 

## 2. Materials and Methods

### 2.1. Insulin HPLC Determination 

All data sets used in this study have been previously published by the authors [14,15] and are available as Appendix A. However, the analysis of samples has been described in detail in order to determine the uncertainty of measured concentrations to facilitate the interpretation and subsequent data analysis.

#### 2.1.1. Analysis of Samples

The sample treatment was as follows: Pure insulin samples were prepared by direct dilution with 0.05 M HCl over a range of concentrations of 2–8 µg/mL and analyzed the same day. In the case of pharmaceutical preparation, 100 µL homogeneous sample was withdrawn from vials and diluted with 0.05 M HCl to 10 mL flask, and finally, 1.0 mL of this solution was half-diluted with the same solvent to obtain concentration values within the calibration range. Unless otherwise indicated, all samples were analyzed in triplicate.

In order to validate the analytical method, seven standard solutions were prepared using human insulin (Batch: H01003, Novo Biolabs, Madrid, Spain) at concentrations of 2–8 µg/mL. Each sample was replicated four times. The analysis of variance (ANOVA) of linear regression confirms the linearity of the method used through the rejection of the null hypothesis of deviation of the linearity for a significative level of 0.05 (α = 0.05).

In order to calibrate the RP-HPLC system, an insulin solution sample was analyzed on a daily basis, as standard. Insulin solution, used as standard, was stored between 2 and 8 °C for two years. During this time, 144 samples from seven batches were analyzed [14]. The standard was analyzed every workday to determine the peak area. 

#### 2.1.2. Long-Term Stability Study

Six batches of the same pharmaceutical preparation (insulin solution, Actrapid^®^ 40 UI/mL, Novo Nordisk, Madrid, Spain) were divided into two groups. Group 1 (batches 1, 2, and 4) was used to estimate the shelf life following the sampling time proposed in the Tripartite Guideline [16] (0, 3, 6, 9, and 12 months). The samples were stored protected from light at a thermostatically controlled room temperature for two years (22.7 ± 1.6 °C). Group 2 (batches 3, 5, and 6) was used to test the proposed procedure for setting release limits. These batches were characterized by a lower number of sampling time points. See the Appendix A for further details.

### 2.2. Process Capacity Indices

The Cpk capability index is defined by the equation:(1)$$C_{pk}=min\left(\frac{USL-\mu }{3\sigma },\frac{\mu -LSL}{3\sigma }\right)=C_{p}\left(1-K\right)$$
where [LSL, USL] is the specification interval, μ is the process mean, and σ is the process standard deviation. The Cp capability index and K factor are defined by:(2)$$C_{p}=\frac{USL-LSL}{6\sigma }$$
(3)$$K=\frac{(USL+LSL)/2-\mu }{USL-LSL}$$

Capacity indices are related to defects per million opportunities (DPMO) by the equation:(4)$$DPMO=10^{6}\frac{c}{n\;x\;n_{CTDQ}}$$

For a variable with distribution N (μ, σ^2^), the relationship between DPMO and process capability is the parameter σ-level [17]. The starting point is a 6σ-level process for which the width of the acceptability interval, USL − LSL, is equal to six standard deviations, namely, USL − LSL = 6 × σ. By convention, DPMO is calculated when the mean of the process is displaced by 1.5 × σ, and thus:(5)$$DPMO=10^{6}\int_{USL}^{\infty }\phi (x\left|\mu +1.5\sigma ,\sigma \right)dx$$

A process with a 6σ level has a Cp capability index equal to 2, which is considered a high-quality standard [17]. When the process is not centered in the specification interval, Cpk is used instead of Cp. The relationship between Cpk and the proportion of defective units, i.e., measurements outside the specification interval, are performed by the equation [18]:(6)$$2\Phi \left(3C_{pk}\right)-1\leq 1-f_{D}\leq \Phi (3C_{pk})$$

The upper value corresponds to a centered process, i.e., when Cpk = Cp (see Table 1). To overcome problems related to the computation of Cpk under the batch effect [19], this was computed from the estimated parameters of the mixed effect model:(7)$$y_{ij}=\mu +\alpha_{i}+\epsilon_{ij}$$
where μ is the fixed effect population mean, α_i_, i = 1 k the random effect due to batch i, and εij is the random effect of j observation (j = 1 nk) from batch i. The random effects are assumed to be mutually independent with distributions α~N (0, ${\hat{\sigma }}_{\alpha }^{2}$) and ε~N (0,${\hat{\sigma }}_{\varepsilon }^{2}$). The σ in Equation (1) is then substituted by $\left({\hat{\sigma }}_{\alpha }^{2}+{\hat{\sigma }}_{\varepsilon }^{2}\right)$ and μ by $\hat{\mu }$. Model parameters were estimated using the function lme () from library nlme [20], and the confidence intervals for Cpk were obtained using the function bootstrap () from lmersampler library [21]. All packages are available in the free R-program (www.r-project.org). 

### 2.3. Stability Data Analysis

Stability data analysis was performed assuming a linear decrease of insulin concentration with time using the statistical model:(8)$$y_{i,j}=\left(\alpha_{0}+\alpha_{i}\right)+\left(\beta_{0}+\beta_{i}\right)t_{ij}+\epsilon_{ij}$$
where y_ij_ is the j observation (j = 1, … nj) made on batch i (i = 1 … k) at time t_ij_, α_0_ and β_0_ are the average of intercept and slope, respectively, and αi and βi are the batch i contributions to intercept and slope, respectively; ε_ij_ is the observation error assumed ε~N (0, σ^2^_ε_ I). Model parameters were estimated with the lm() function from R application. Sum zero restriction ∑αi = 0 and ∑βi = 0 for i = 1 to k was applied using the optional argument in function lm () contrasts = list (batch = “contr.sum”). This restriction is required when function Anova () from library car [22] is applied to conduct type III analysis of variance of the regression model to test the poolability of the results from the three assayed batches. Critical level is computed using the method proposed by Ruberg and Stegeman [23]. 

## 3. Results and Discussion

As mentioned above, drug product shelf life depends on the true stability of the drug product, performance of the manufacturing process, and analytical method uncertainty, as well as on the experimental design and the statistical model used for stability data interpretation. Thus, it is pertinent to answer the following questions when analyzing drug stability data: firstly, the uncertainty of the analytical method regarding the range of drug concentration in the stability test; secondly, the capability of the manufacturing process; thirdly, the poolability criteria for data from different batches; fourthly, how to set the LRL and a suitable statistical method to check new batches.

### 3.1. HPLC Validation and Uncertainty Determination

The HPLC analytical method used in this article was previously published [9]. In short, linear calibration was conducted with four independent assays, and in each assay, seven independent samples with insulin concentration ranges from 2 to 8 µg/mL were prepared. The statistical model included concentration, day (as a factor), and the interaction concentration by day coefficients. The null hypothesis about day and interaction concentration by day was accepted, and therefore a simple linear model was used. Coefficient estimates were: intercept (b0) equal to −5.936, and slope (b1) equal to 9.294. Adjusted R^2^ was equal to 99.77%, and the standard deviation of residuals was equal to 0.9029. Uncertainties of measured concentrations were obtained by applying the delta method to the equation:(9)$$x_{0}=\frac{y_{i}-b_{0}}{b_{1}}f$$
with yi being a new HPLC area measurement (µV x s), x_0_ the resulting concentration (µg x mL), and f a factor due to the sample preparation; in the case here, it is equal to 0.2. Insulin concentration calculated for the mean value of HPLC area obtained in the process capability analysis (see below), equal to 62.0629 (µV x s), was 1.463 µg/mL, and its uncertainty expressed as one standard deviation obtained by the delta method was 0.02023, namely, a variation coefficient equal to 1.38%. This data is within the interval usually encountered in HPLC drug composition determination, but due to the short range of drug concentration in the stability testing, usually in the range 90–100% of the labeled drug concentration and there will be a reduction in the precision of the shelf life estimation.

### 3.2. Process Capability

Table 2 summarizes the sample statistics for each batch: number of samples assayed at time zero (n), minimum (min) and maximum (max) observed insulin concentrations, mean (mean), and standard deviation (sd), as well as the natural estimation of Cpk. As mentioned above, the six batches were divided into two groups. Group 1 (batches 1, 2, and 4) includes the batches used to estimate the shelf life; Group 2 (batches 3, 5, and 6) was used to test the method for setting the LRL. Figure 1 left shows the box plot of the observations by batch together with LSL and USL, equal to 95 and 105% of the labeled insulin concentration, respectively; LCL and UCL refer to the lower and upper capability limits, respectively, based on the µ ± 3 σ rule and computed as will be seen later; unlabeled dotted lines are the labeled insulin concentration (1.454 µg/mL) and the mean insulin concentration estimated from the process capability analysis (1.462 µg/mL; see Table 3). Mean average values for the six batches were above the target value, a fact that, after deep data revision, the authors were unable to explain. Cpk values above 2.0 and box plot show that the manufacturing process complies with the quality standards and that it is under statistical quality control. Figure 1-right shows the probability plot to assess the normal distribution of data.

Cpk, as well other capability indices, shows several advantages in process capability analysis. Firstly, it provides a universally interpretable measurement of the process capability through σ level and defective proportion; secondly, sample statistics for natural estimates of capability indices are suitable for hypothesis testing [24], statistical quality control by acceptance sampling [25], or combining with control charts in process monitoring [26]. In the authors’ opinion, two more issues must be attended to when capability indices are used to monitor process performance: the effect of the uncertainty of the analytical methods and the batch effect mentioned above. Analytical method uncertainty could be an important source of variability, if not the only one, when the drug product is an homogeneous system; for example, when it is a solution, and the differences between samples from the same batch are expected to be null. When both process and analytical uncertainty play significant roles, the statistics of the capability index are more complex [27].

In order to overcome the role played by analytical uncertainty and batch effect, as well as the estimation of variance components in order to set the release limits, the authors chose the Cpk calculation through the estimation of the parameters of the statistical model depicted in Equation (7) using data from batches 1, 2, and 4. Table 3 shows the estimates of the statistical model parameters as well as their two-sided 95% confidence intervals. As mentioned above, the 95% confidence interval for mean insulin concentration is above the labeled insulin concentration; on the other hand, between-batch standard deviation (0.0015891) is smaller than within-batch standard deviation (0.0087035), an extra argument to conclude that the manufacturing process is under statistical quality control. Moreover, ${\hat{C}}_{pk}$ estimated by the bootstrap method is equal to 2.42, and its 95% confidence interval is equal to [1.97–2.83]. These data confirm that the process is not only under statistical quality control, but also that it complies with the specification regarding drug content, as it exceeded the 4-σ level which is customarily used in drug product pharmaceutical manufacturing. In conclusion, the data from Group 1 batches are suitable for expiry time estimation.

### 3.3. Stability

Table 4 summarizes stability results by batch as well as for the poolability data: intercept (b0), slope (b1), and residual standard deviation (RSD), as well as shelf lives calculated following the ICH guideline (T_ICH_); as can be seen, similar results were obtained for the three batches. Table 5 shows ANCOVA type III for poolability data. The null hypothesis for the origin of variation times x batch was accepted for α level equal to 0.25, as required by the ICH guideline. To test that this result does not arise from the lack of power of the regression analysis, the critical alpha level was computed using the Ruberg and Stegeman method [23] for batch poolability data using as alternative to null hypothesis of equal slopes:$$Ha:\beta_{1}=0,\;\;\;\;\;\beta_{2}=0,\beta_{3}=∆$$
where ∆ was set equal to 2% of the LSL (95% of the target concentration) above the proposed shelf life, eight months (∆ = −8.606 × 10^−3^, equal to 2% of the LSL over 4.1 months); type II error was set equal to 0.10. The critical significance level was equal to 0.002, lower than the *p* value obtained in ANOVA type III for the interaction term batch x times, 0.46 (see Table 5). Therefore, the authors concluded that the criteria for batch poolability have been met.

Figure 2 graphically shows the stability results together with the results from the process capability. This plot shows the experimental points for the three batches, and the fitted line (solid line), the one-sided lower limit of the 95% confidence interval for poolability data (bold dashed line), i.e.:(10)$$\hat{y}-t_{1-\alpha ,n-2}\cdot s\sqrt{\frac{1}{n}+\frac{{\left(x_{i}-\overset{-}{x}\right)}^{2}}{S_{xx}}}$$
as well as the one-sided lower limit of the 95% confidence interval for a future individual reportable value:(11)$$\hat{y}-t_{1-\alpha ,n-2}\cdot s\sqrt{1+\frac{1}{n}+\frac{{\left(x_{i}-\overset{-}{x}\right)}^{2}}{S_{xx}}}$$

The authors unsuccessfully tried to perform the parameter estimation for the mixed effect model using the lme function from nlme package [20], which failed in the optimization convergence, as well as the proposed method by Murphy and Hofer [2] based on the interpretation of ANCOVA assuming random slopes. In this case, the expected mean square for the interaction term batch x times is equal to $\sigma^{2}+R\sigma_{\beta }^{2}$, $R=(W-\frac{\sigma_{i}^{2}}{W})/(k-1)$, $w_{i}=\sum_{j}{\left(x_{ij}-{\overset{-}{x}}_{i}\right)}^{2}$, and $W=\sum_{i}w_{i}$. This method gives an unbiased estimation for $\sigma_{\beta }^{2}$ only if the data are balanced, i.e., the same number of points for all the batches. Another drawback of this method is the high probability of obtaining negative estimates for $\sigma_{\beta }^{2}$ when $\sigma^{2}>\sigma_{\beta }^{2}$, as happened in the present study. 

The method outlined by the ICH guideline to set the shelf life is the intersection of the curve defined by Equation (10) with the LSL. This method makes sense when within- batch variability is expected to be negligible, as happens in drug solutions, and the main source of variability is the uncertainty of the analytical method. Under these circumstances, the ICH method warrants the quality for the entire batch. However, when within-batch variability is expected to be substantial, as would happen for tablets with a low drug content, or in freeze-dried drug products which show within-batch differences in drug degradation due to the residual moisture content [28], a large proportion of the units will have a drug content below the LSL, because the width of the confidence interval tends to zero when $x_{i}\approx \overset{-}{x}$ and n is large. Thus, when substantial within-batch variability is expected, it makes sense to calculate the shelf life as the intersection of the one-sided lower limit of the 95% confidence interval for a future individual reportable value with the lower specification limit, as already suggested by other authors such as Wei [9]. Figure 2 includes both values, T_ICH_, equal to 8.73 months, and the value based on prediction interval, 7 months.

### 3.4. Release Limits

In order to assure that the drug concentration at the end of shelf life is above the LSL, the release interval, [LRL, URL], must be within the interval defined by the process capability, i.e., [LCL, UCL] ∈ [LRL, URL]. Otherwise, the process capability would be incapable of complying with the drug product specification at the expiry time. The authors suggest a possible solution to this problem by setting LRL = LCL, and they estimate the time for which y_n+1,T_ = LSL. In this way, testing drug content for batch release will be performed assuring that drug manufacturing is under statistical quality control and meets drug product specification in agreement with the quality by design principles. 

The ADG method can be derived from a stability statement similar to the conditioned rule proposed by Wei [9]:(12)$$P\left(y_{b+1,T}<LSL\right)=\alpha$$

Let
(13)$$y_{n+1,T}=y_{0}+\beta \cdot T+\epsilon$$
where y_0_ is the true but unknown batch mean content at t = 0, β is the degradation rate, T is the prescribed shelf life, and Є is the error with distribution N~ (0, σ*^2^*). Setting $y_{n+1,T}=LSL$, the expected value of the difference LSL-y*_o_*-b·T is equal to zero and its variance is equal to $T^{2}\sigma_{b}^{2}+\sigma^{2}$, and therefore the ratio of both terms is: (14)$$\frac{LSL-y_{0}-b\cdot T}{\sqrt{T^{2}+\sigma_{b}^{2}+\sigma^{2}}}$$

This ratio approximates the t-student distribution. Resolving for y_0_, now equal to LRL, we obtain the ADG equation:(15)$${LRL}_{ADG}=LSL-bT-t_{1-\alpha ,\nu }\sqrt{T^{2}s_{b}^{2}+s^{2}}$$

The slope is estimated from poolability stability data; s^2^_b_ is the slope variance extracted from the covariance matrix $s_{r}^{2}(X^{T}{X)}^{-1}$, with s^2^_r_ being the variance of the residuals; s^2^ in the above equation is the variance of a new observation made in a new batch at time equal to 0, estimated from the process capability analysis. 

The original ADG method, as well other ones intended to calculate the LRL, could give results even above the labeled concentration of the drug product. In such cases it is necessary to try a new and shorter value for the shelf life. Instead, we proposed setting the value of LRL equal to the lower limit of the confidence interval derived from process capacity analysis, and then computing T from the last equation using bisection method. In this context, LRL_ADG_ is an increasing function of T, as shown in Figure 3, where LRL_ADG_ is plotted for shelf lives ranging from 1 to 8 months. Some tradeoff between T, LRL_ADG_, and process capability is necessary, and with this objective the authors propose setting LRL_ADG_ = LCL and solving for T using the bisection method implemented in the bisection () function from the cmna package [29]. The resulting expiry time was equal to 4.1 months. This result is far less than the expiry time using the ICH method, 8.73 months, but it is the price to pay for controlling the batch defective fraction at the expiry time considering the process capability. 

Once the release criteria [LRL, URL] = [LCL, UCL] have been set, some statistical method is necessary to test that the new batch meets the specifications. There are several methods, but one suitable for samples of low to medium sizes is the ASTM E2709-12 [30]. Following this method, the release criterion is LRL ≤ yi ≤ URL for i = 1 … n, where yi are the observations from a new batch and n is the sample size. The ASTM E2709-12 procedure leads us to compute a given acceptance limit table. For each value of the sample mean, the maximum value of the standard deviation that would meet the criterion; alternatively, the acceptance table can be shown as a chart depicting the acceptance region. Figure 4 shows the acceptance region for 90% confidence interval 95% coverage, for samples of sizes 10 (inner curve) and 30, and release limits [1.435, 1.488]. Points drawn as circles correspond to the mean and standard deviation of the batches used for process capability analysis (Group 1 in Table 2), and the points drawn as triangles correspond to the batches used to test the procedure (Group 2 in Table 2). The two conclusions that can be drawn from this figure are: firstly, there is no real advantage of using samples of size 30 instead of samples of size 10; secondly, batches from Group 2 comply with the release criteria.

The analysis of the outlined procedure provides some clues to improve the results. One obvious one is to enhance the manufacturing process and to reduce the uncertainty of the analytical method. The results obtained in the present study show that the process capability, including the uncertainty of the analytical method, is quite satisfactory regarding the specification limits used in stability studies, ±5% if the drug would not be undergoing a degradation process, as can be concluded from the Cpk values. However, LCL, equal to 1.427 mg/mL, is 98.17% of the labeled drug concentration, reducing the room for drug degradation to approximately 3%. Another issue is the sampling times. ICH guidelines recommend taking samples at times 0, 3, 6, 9, 12, 18, and 24 months, and then every 12 months until study finalization. Samples are usually collected until drug concentration is just below the LSL, but as:(16)$$s_{b1}^{2}=\frac{s_{r}^{2}}{\sum {\left(x_{i}-\overset{-}{x}\right)}^{2}}$$
$s_{b}^{2}$ could be lower, increasing the number of sampling times at the end of the experiment.

Another advantage of the procedure outlined in this paper is that it is not based on the correlation between observations made at time zero and those made at the expiry time. For a manufacturing process under statistical quality control, it is expected that between- batches variance would be far less than within-batch variance, and the intraclass correlation coefficient will approach to zero; from data in Table 3 we obtain $\hat{\rho }$ = 0.0433. As pointed out by Chen and van der Vaart [7], setting a limit at time zero to control stability at time T is practically impossible.

## 4. Conclusions

The method reported in the present paper is based on two premises: firstly, batches used for testing drug stability, even if they are pilot plant ones, must be manufactured under quality by design principles, and the manufacturing process must be under statistical quality control; secondly, the expiry time is the result of a tradeoff between process capability, drug stability, and analytical method uncertainty. In order to do this, the authors propose calculating the expiry time from the LCL derived from process capability analysis and from the degradation rate, applying a modified version of the proposed method by Allen et al. [6] This approach favors batch acceptance over the length of the shelf life, reducing the patient’s risk.

## Figures and Tables

**Figure 1 pharmaceutics-15-01070-f001:**
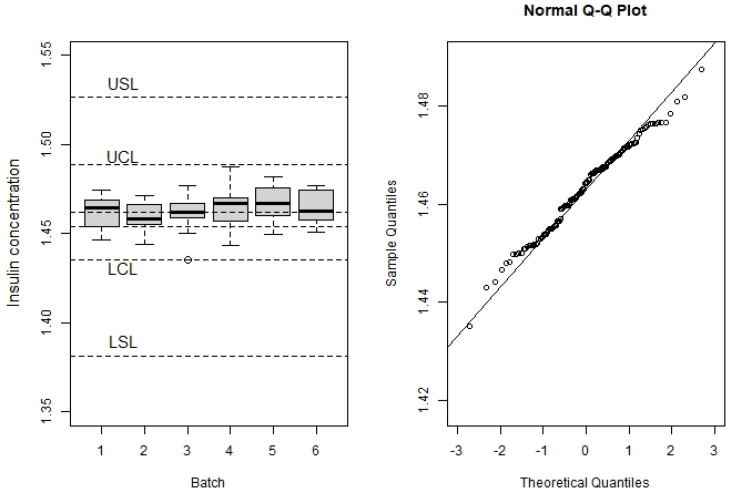
The box plot of the observations by batch, together with the lower and upper specification limits and capabilities (**left**), and Normal Q-Q plot (**right**).

**Figure 2 pharmaceutics-15-01070-f002:**
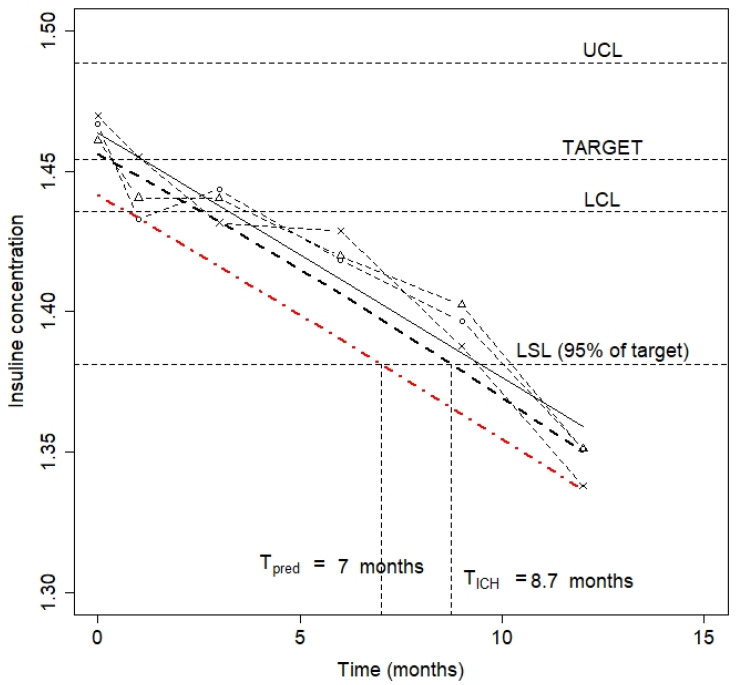
Stability profile.

**Figure 3 pharmaceutics-15-01070-f003:**
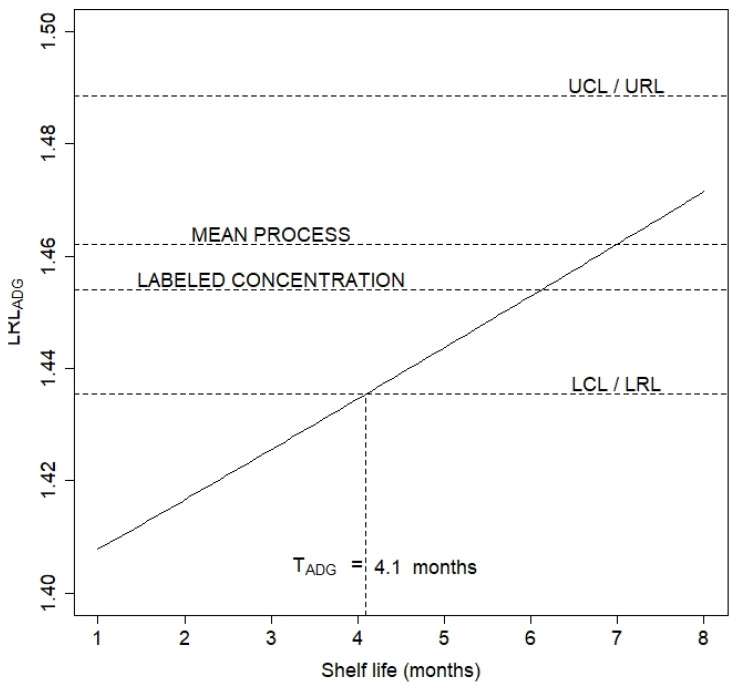
Lower release limit versus expiry time (TE).

**Figure 4 pharmaceutics-15-01070-f004:**
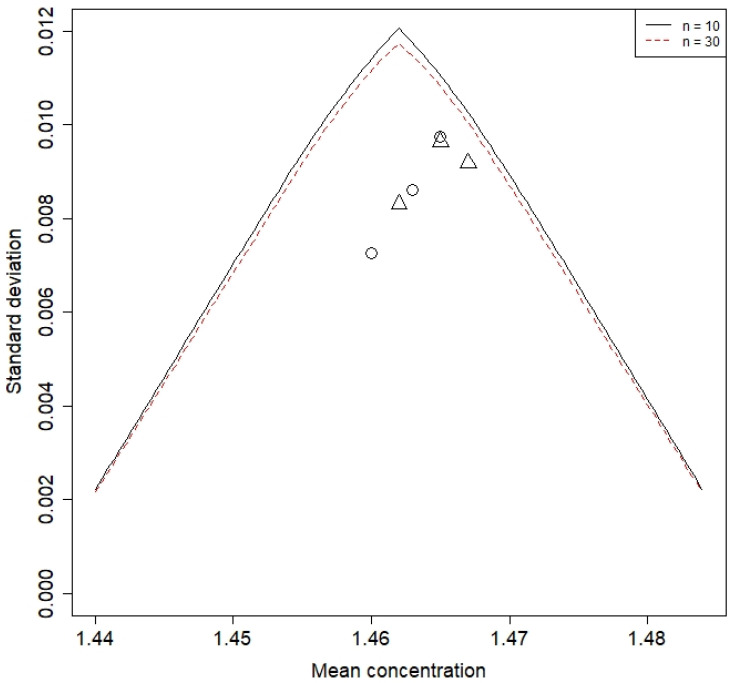
Release criterion acceptance region (Triangle corresponds to Group 2 batches; circle corresponds to Group 1 batches).

**Table 1 pharmaceutics-15-01070-t001:** Cpk values and defective proportions.

Sigma Level	Cpk	Lower	Upper
3	1.00	2.700 × 10^−3^	1.350 × 10^−3^
4	1.33	6.607 × 10^−5^	3.304 × 10^−5^
5	1.66	6.358 × 10^−7^	3.179 × 10^−7^
6	2.00	1.973 × 10^−9^	9.866 × 10^−10^

**Table 2 pharmaceutics-15-01070-t002:** Batch descriptive statistics.

Group 1	n	min	max	mean	sd	Cpk
#1	22	1.447	1.474	1.463	8.602 × 10^−3^	2.49
#2	22	1.444	1.471	1.460	7.256 × 10^−3^	3.07
#4	28	1.443	1.488	1.465	9.757 × 10^−3^	2.12
Group 2						
#3	34	1.435	1.477	1.462	8.311 × 10^−3^	2.61
#5	30	1.450	1.482	1.467	9.193 × 10^−3^	2.17
#6	8	1.451	1.477	1.465	9.639 × 10^−3^	2.14

**Table 3 pharmaceutics-15-01070-t003:** Model parameter (Equation (7)) for process capability assessment.

Parameter	Lower	Estimate	Upper
Intercept	1.4597	1.4625	1.4652
Batch sd	1.80 × 10^−4^	1.59 × 10^−3^	1.41 × 10^−2^
Residual sd	7.37 × 10^−3^	8.70 × 10^−3^	1.03 × 10^−2^

**Table 4 pharmaceutics-15-01070-t004:** Results of stability study by individual and poolability batches (T_ICH_ in months).

Batch	b0	b1	RSD	T_ICH_
1	1.461	−8.204 × 10^−3^	1.360 × 10^−2^	7.95
2	1.460	−7.939 × 10^−3^	1.210 × 10^−2^	8.30
4	1.470	−9.980 × 10^−3^	1.275 × 10^−2^	7.61
Poolability	1.464	−8.708 × 10^−3^	1.186 × 10^−2^	8.73

**Table 5 pharmaceutics-15-01070-t005:** ANCOVA type III sum of squares.

Source of Variation	Sum of Square	DF	Mean Square	F	Pr (>F)
Intercept	5.235	1	5.235	31808	0.000
Times	7.459 × 10^−3^	1	7.459 × 10^−3^	45.319	0.000
Batch	1.507 × 10^−4^	2	7.536 × 10^−5^	0.458	0.643
Times × Batch	2.730 × 10^−4^	2	1.365 × 10^−4^	0.829	0.460
Residuals	1.975 × 10^−3^	12	1.646 × 10^−4^		

## Data Availability

Data are available in this article and in the associated Appendix A.

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
