# Peer review of "Drug Shelf Life and Release Limits Estimation Based on Manufacturing Process Capability"

_pharmaceutics, 2023, doi:10.3390/pharmaceutics15041070_

Round 1

Reviewer 1 Report

The manuscript is well written, and the authors proposed a method to set the release limits based on the manufacturing process capability and degradation rate using a modified version of the conventional method. Authors are advised to revise the manuscript following the comment given below.

1.      Release of manufacturing lots and maintaining the quality to satisfy customers' demand is a challenging task, but this manuscript is not the first report on the same. Therefore, authors are advised to change the word "novel" in the last line of the abstract.

2.      Introduction/ line-2: It is not meaningful. What to establish?

3.      Line 35: Several brackets were used to explain LSL and USL, which can be avoided. Similarly, in line 36. Straightway authors can write the full form of the abbreviations at their first use.

4.      Line 40: What does this notation indicate?

5.      Authors can speak about the limitations of Frequentist and Bayesian approaches. A few limitations of Bayesian are there, but the other method is not there. In the same paragraph, authors are advised to update the research information on the same. Because references 1-7 are very old except the number 6. Then it will be easy to introduce the objective of this work.

6.      Why have authors prepared 7 standard solutions? Is it to maintain statistical significance?

7.      Table-1: The lower and upper limits should be written fully on decimal, or a multiplication sign should be given before the 10th power. Same for other tables too.

8.      Line 167: What is yi? What is exactly the f factor? Why is it 0.2?

9.      The conclusion is very short. Authors can bring some result-based statements to conclude and establish their observations.

10.  Decision: Minor corrections. 

Author Response

REPLIES TO REVIEWERS´ COMMENTS

Reviewer #1

The manuscript is well written, and the authors proposed a method to set the release limits based on the manufacturing process capability and degradation rate using a modified version of the conventional method. Authors are advised to revise the manuscript following the comment given below.

  1. Release of manufacturing lots and maintaining the quality to satisfy customers' demand is a challenging task, but this manuscript is not the first report on the same. Therefore, authors are advised to change the word "novel" in the last line of the abstract.

The phrase “To the best of the authors’ knowledge, this is a novel procedure to estimate LRL” could be confusing because it is not clear to which procedure it refers. This line was deleted in the abstract and at the end of the introduction.

  1. Introduction/ line-2: It is not meaningful. What to establish?

This sentence is a copy of the general principles set out in the ICH Q 1E Guideline (see Section 2.1) where the purpose of a stability study is “to establish, based on testing a minimum of three batches of the drug substance or product, a retest period or shelf life and label storage instructions applicable to all future batches manufactured and packaged under similar circumstances”. Certainly, the phrase could be modified, but in our opinion, the use of the original text is more appropriate.

  1. Line 35: Several brackets were used to explain LSL and USL, which can be avoided. Similarly, in line 36. Straightway authors can write the full form of the abbreviations at their first use.

Thank you for your suggestions. The brackets were deleted and we write the full form of the abbreviations at their first use.

  1. Line 40: What does this notation indicate?

There is a mistaken in the notation. The correct formula is the following: y Є [LRL, URL], being “y” an observation from a future batch. We have changed the text.

  1. Authors can speak about the limitations of Frequentist and Bayesian approaches. A few limitations of Bayesian are there, but the other method is not there. In the same paragraph, authors are advised to update the research information on the same. Because references 1-7 are very old except the number 6. Then it will be easy to introduce the objective of this work.

It is true that some of the references are older, but we believe that all of the references used in this section are relevant to the release limit determination procedure and the objectives of our work. In addition, the works published by Allen et al (1991) and Wei, G.C. (1998) were pioneers in the determination of drug release limits.

It is true, the review of both frequentist and Bayesian methods is very short; it is limited to the paragraph in lines 52 – 62 lines. But we have some reasons:

  1. a) Montes et al., (2019) and Chen and van der Vaart (2022) (references 6 and 5 in our paper) review both types of methods, and we think it is better to avoid repeating their arguments.
  2. b) The oldest reference, Allen et al. (1991), has been included because the method they propose seems to be the most used in the pharmaceutical industry, as it has been named by Chen and van der Vaart.
  3. c) Our paper was motivated by the study of the methods proposed by Montes et al., and later by the methods proposed by Chen and van der Vaart, but soon we realized that these and other methods had been developed from a statistical point of view, without paying attention to the process capability, and started to analyze the outlined method in our paper. On the other hand, the combination of other methods, as those outlined by Montes et al. and Chen and van der Vaart, with process capability require

  1. d) The point that makes the difference between our proposed method and others is that the shelf life results from a compromise between process capability and degradation rate (Figure 3).

  1. Why have authors prepared 7 standard solutions? Is it to maintain statistical significance?

It is correct.

  1. Table-1: The lower and upper limits should be written fully on decimal, or a multiplication sign should be given before the 10th power. Same for other tables too.

Thank you for your suggestion. We used the multiplication sign before the 10th power. All tables were revised.

  1. Line 167: What is yi? What is exactly the f factor? Why is it 0.2?

There was an error in equation 9, which has been corrected. The correct equation is:

where yi is the new measurement of the HPLC peak area, and f the dilution factor, in our case, equal to 0.2

  1. The conclusion is very short. Authors can bring some result-based statements to conclude and establish their observations.

It is true that we can bring some result-based statement to expose our conclusions, but we think the conclusion is very clear and concise since it summarizes the conditions that the batches should meet in the stability study and how to estimate the expire time using the manufacturing process capability data and the degradation rate.

Nevertheless, the Bayesian approach to both process manufacturing control, especially for continuous manufacturing with subsampling, and release limit setting deserves more attention, but these topics are beyond the scope of this paper.

Reviewer 2 Report

This manuscript deals with a novel way to determine the lower release limit of drug batches, aiming to reduce the simulated data.  To that aim, employing the method proposed by Allen, the authors used the manufacturing process capability data, in a process involving 5 steps. 

The overall presented information is valuable. However, I find the manuscript should be presented in a more attractive way in order to raise the reader´s interest.  For instance, I suggest modifying the title, which seems a plain paragraph. It should also be included one graphical abstract. Finally, I suggest the authors broaden the scope of the presentation, in order to explicitly include the nanotechnological pharmaceutical products. In case this should not possible, the authors should also explicitly mention it. 

Author Response

Reviewer #2:

This manuscript deals with a novel way to determine the lower release limit of drug batches, aiming to reduce the simulated data.  To that aim, employing the method proposed by Allen, the authors used the manufacturing process capability data, in a process involving 5 steps.

The overall presented information is valuable. However, I find the manuscript should be presented in a more attractive way in order to raise the reader´s interest.  For instance, I suggest modifying the title, which seems a plain paragraph. It should also be included one graphical abstract. Finally, I suggest the authors broaden the scope of the presentation, in order to explicitly include the nanotechnological pharmaceutical products. In case this should not possible, the authors should also explicitly mention it.

Thank you very much for yours comments about our paper.

New title:

Drug shelf-life and release limits estimation based on manufacturing process capability

We have no experience of the stability of nanotechnological products. The application of our proposed methodology to such products would require the identification of the quality critical attributes, the specification limits of such attributes and the knowledge of how they evolve over time.

In our manuscript we analyzed the simplest case: the critical attribute is the drug concentration; the specifications are well established, and as long as the degraded fraction is not too high, it is expected that the degradation rate could be correctly interpreted with a linear model.

Reviewer 3 Report

Review summary: The topic of release limits and process capability have been discussed in the literature a lot.  There is not much new added in this work to be honest, apart from a slight variation of the "Allen" approach from 1991. What is compelling is the supplemental R code, which adds to transparency and might be of use to the interested reader.

Review: Major recommendation to the authors are:

- State clearly how your variation of the "Allen" approach is different and what is the benefit of this approach. To my understanding, shelf life is not fixed and is reduced to however little needed, to fit the process into the release limits. It should be made clear that with this procedure, shelf life will be reduced for products with poor manufacturing control and that this approach favors batch acceptability over shelf life period. In reality, it may be better to keep tighter release limits and potentially reject batches at release in order to have a useful shelf life. This trade-off should be discussed in the paper.

- Six batches worth of data is presented and these are divided into two groups of three for model building and validation. It is not clear how the batches were assigned into these groups and what would happen with different assignation. It might be more transparent to include a validation in the sense of leave-one-out or a similar non-biased approach.

- The R syntax included could be advertised to the reader, possibly in the abstract

- The topics covered here are a subject of many scientific publications and if an authoritative review or a book chapter exists, it should be cited. There are two self-citations and please assess, if both are required.

Detailed comments:

L15: include a summyr of what is changed from "Allen" approach.

L27: describe why you feel this is controversial.

L46 How is it clear that that designs and analyses are not suitable, explain to the reader.

L61. Explain why

L62-64 This sentence is very unclear and should be fitted with an explanation or a literature reference.

L66-67: Describe a bit more the "Allen" method so the reader does not need to study the original paper. Because the topic of this paper is mostly about the variation of the "Allen" method.

L101 Explain how batches were assigned to each group, try an alternative non-biased validation scenario (eg leave-one-out).

L108: Add lit reference

L166 add units

L170 Should yi read y0?

L175: variation coefficient obtained by th edelta method (1.38 %) shuld be compared to that of the method validation-robustness results and typical RSD from table 2 (which is 0.6% which is somewhat lower than 1.38%). Explain in the paper how that relates to RSD from table 4.

L225 in the introduction. six sigma is stated; here 4 sigma is mentioned.

L240 -241. Please explain into more detail how value -8.606e-03 was obtained for delta.

Figure 2: If you added "95% of target) to the LSL horizontal line, it would increase clarity.

L285-286: Is it not the other way around? (process capability should lie within the release interval)

L286-287: process capability relates to values at release and cannot relate to value at shelf life, this sentence needs more clarity.

L307 I think this is a good place to include a discussion into what is more appropriate : to reject batches at release due to a tighter release limit and longer shelf life or to decrease the shelf life in order to keep all batches within release limit.

L311: Should yi read mean(yi)?

Figure 4: can you make the inner line for n=10 dashed for clarity and include a legend?

L359: Add to the conclusion that this approach favors batch acceptance over the length of shelf life (the former being manufacturer-centric and the latter being consumer-preferred) 

Author Response

Manuscript ID: Pharmaceutics_2237290 revised version

March, 23th, 2023

Dear Ms. Nonna Li

Thank you very much for your suggestions about our paper. We included most of the suggestions given by the reviewer #3. The changes made in the revised manuscript are marked up using the “Track Changes” function from MS Word. Please find enclosed a detailed list of our responses to commentaries from the reviewer.

I look forward to hearing from you,

    Yours Sincerely,

Alexis Oliva

REPLIES TO REVIEWERS´ COMMENTS

Reviewer #3

Review summary: The topic of release limits and process capability have been discussed in the literature a lot.  There is not much new added in this work to be honest, apart from a slight variation of the "Allen" approach from 1991. What is compelling is the supplemental R code, which adds to transparency and might be of use to the interested reader.

Our point of view is that assessing of process capability prior to stability testing is need by two reasons.

First, statistical test for pooling batches in stability studies are not suitable to test differences in drug concentration (or any other drug product property) at zero time because lack of statistical power, as well as because the correlation among the estimates of the intercept and slope. Second, if the manufacture process is not under statistical control and meets the drug product specification, do not make sense to extrapolate the assessed shelf life to future batches. Only Bayesian methods assume, at least in an implicit way, that the process manufacture is under statistical quality control, but this do not mean that the specifications of the drug product are meet.

From the statistical analysis point of view, our work does not add any novelty. As we stated in the last paragraph of our paper, the aim of this work is to take into account the process capacity analysis of the drug product manufacture into stability studies to set the release limit.

Review: Major recommendation to the authors are:

- State clearly how your variation of the "Allen" approach is different and what is the benefit of this approach. To my understanding, shelf life is not fixed and is reduced to however little needed, to fit the process into the release limits. It should be made clear that with this procedure, shelf life will be reduced for products with poor manufacturing control and that this approach favors batch acceptability over shelf life period. In reality, it may be better to keep tighter release limits and potentially reject batches at release in order to have a useful shelf life. This trade-off should be discussed in the paper.

Your interpretation of our proposal is right. We do not state a hypothesis about shelf life and then get the estimation of the release limit; instead, we set the release limit from process capability and then we estimate the shelf life. But we discrepe about what is a “poor manufacturing control”. Overall variability around 1% (standard deviation as percent of initial concentration), accounting both for both the analytical uncertainty and process variability, can be reached only for homogeneous systems (i.e. dissolutions). If the specification limits are, for example 95 - 105 %, overall variability equal to 2% will limit substantially the drug product shelf life.

- Six batches worth of data is presented and these are divided into two groups of three for model building and validation. It is not clear how the batches were assigned into these groups and what would happen with different assignation. It might be more transparent to include a validation in the sense of leave-one-out or a similar non-biased approach.

The procedure to divide the batches in two groups has been added. Only batches 1, 2 and 4 were used in a extended stability program.

- The R syntax included could be advertised to the reader, possibly in the abstract

This suggestion was included in the abstract.

- The topics covered here are a subject of many scientific publications and if an authoritative review or a book chapter exists, it should be cited. There are two self-citations and please assess, if both are required.

Thank you for your suggestions. However, we have included both self-citations because they describe the original data and methods.

Detailed comments:

L15: include a summary of what is changed from "Allen" approach.

A brief summary was included in the abstract

L27: describe why you feel this is controversial and L46: How is it clear that that designs and analyses are not suitable, explain to the reader.

It is well know that experimental design especified by ICH guidelines (3 batches, a small number of sampling points) are defficient from a statistical point of view.

L61. Explain why

Two reasons:

 – Prior distribution for Bayesian methods needs to get information from many batches, an information that it is not available when a new drug product is under development.

 – Prior distribution are useful only if it fulfill the drug product specifications.  Bayesian method would requiere a wider analysis, out of the scope of our work.

L66-67: Describe a bit more the "Allen" method so the reader does not need to study the original paper. Because the topic of this paper is mostly about the variation of the "Allen" method.

Allen et al. method is now described in detail in the discussion.

L101 Explain how batches were assigned to each group, try an alternative non-biased validation scenario (eg leave-one-out).

The procedure to divide the batches in two groups has been added.

L108: Add lit reference

The following reference has been added. Wu, C et al., Int. J. Production Economics 117 (2009) 338-359

L166 add units

The units for different variables are available in the original publication. However, the units were included in the text.

Peak area = (µV x s); C = (µg/mL); b0 = (µV x s); b1 = (µV x s)/ (µg/mL)

L170 Should yi read y0?

Yes. The equation was modified.

L175: variation coefficient obtained by the delta method (1.38 %) should be compared to that of the method validation-robustness results and typical RSD from table 2 (which is 0.6% which is somewhat lower than 1.38%). Explain in the paper how that relates to RSD from table 4.

L225 in the introduction. six sigma is stated; here 4 sigma is mentioned.

It is true. Six-sigma is the criteria adopted usually to introduce the capacity indices. Hovewer, a sigma level equal to 6 is too stringent sometimes in drug product manufacturing process, and a sigma level equal to 4 use to be accepted as an estandar in the pharmaceutical industry.

L240 -241. Please explain into more detail how value -8.606e-03 was obtained for delta.

This have been explained in the text

Figure 2: If you added "95% of target) to the LSL horizontal line, it would increase clarity.

The figure 2 has been improved. The legend “95% of the target was included according to your recommendation.

L285-286: Is it not the other way around? (process capability should lie within the release interval)

First step in the quality by design of a pharmaceutical drug product is to set the specifications. For a drug product with limited stability, it make sense to set “a priori” the release limits. However, once the manufacture process has been validated, its performance is determined by the interval [LCL, UCL]. The maximum value for the release limit, which corresponds to the larger shelf life, is equal to the lower control limit.  

L286-287: process capability relates to values at release and cannot relate to value at shelf life, this sentence needs more clarity.

We have explained in detail the ADG method; we hope that now our reasoning is best understandable.

L307 I think this is a good place to include a discussion into what is more appropriate: to reject batches at release due to a tighter release limit and longer shelf life or to decrease the shelf life in order to keep all batches within release limit.

Quality by design and pharmaceutial quality system manufacturing are intended to warrants that the drug product will meet the specifications, including “in house” tests as can be the release limits. Any deviation from process manufacturing parameter must be detected in real-time, for example, using control charts; testing final products must have only a confirmatory value.

L311: Should yi read mean(yi)?

Yes.

Figure 4: can you make the inner line for n=10 dashed for clarity and include a legend?

The figure 4 has been improved.

L359: Add to the conclusion that this approach favors batch acceptance over the length of shelf life (the former being manufacturer-centric and the latter being consumer-preferred)

We have included your comment.
